# Investigation of anti-proliferative and anti-angiogenic properties of *Parkia javanica* bark and fruit extracts in zebrafish

**Rasik Dhakal**[1], **Krithika Kalladka**[1], **Achinta Singha**[2], **Dechamma Pandyanda Nanjappa**[1], **Jeshma Ravindra**[1], **Rajeshwari Vittal**[1], **Samir Kumar Sil**[2]\*, **Anirban Chakraborty**[1], **Gunimala Chakraborty**[1]\*

1 Division of Molecular Genetics and Cancer, Nitte University Centre for Science Education and Research, Nitte (Deemed to be University), Mangalore, India, 2 Cell Physiology and Cancer Biology Lab, Department of Human Physiology, Tripura University, Agartala, India

\* gunimala@nitte.edu.in (GC); s_k_sil@yahoo.com (SKS)

**Data Availability Statement:** All relevant data are within the paper.

## Abstract

The use of herbal products as traditional medicines has been a practice in India for centuries. Due to high ethnic diversity, the pool of herbal medicines is enormous, and they are often preferred over modern medicines in certain parts of the country. Cancer is one of the major non-communicable diseases affecting people worldwide. Despite considerable research, cancer is a disease that is still not understood completely, and there have been constant efforts towards the identification of novel drugs or approaches in cancer management. *Parkia javanica*, an important medicinal plant and a rich source of flavonoids and terpenoids, is widely studied for its antioxidant and anti-inflammatory activities. Traditionally, the fruit and bark extracts of *P. javanica* find use as home remedy for dysentery and piles in NE India. Moreover, the fruits are consumed by the people of North-East (NE) India as vegetables, either in steamed or cooked form. In this study, crude extracts of *P. javanica* fruit and bark were obtained, the sub-lethal dose was determined and were then analyzed for anti-proliferative and anti-angiogenic properties using a battery of assays in zebrafish embryos. The sub-lethal concentration 50 ($LC_{50}$) was found to be 28.66 mg/L and 346.66 mg/L for bark and fruit extract respectively, indicating a decreased toxicity of the fruit extract compared to that of the bark. The anti-proliferative and anti-angiogenic properties were more pronounced for the fruit extract compared to the bark extract. Although preliminary, the results of the study suggest that *P. javanica* fruits possess potent anti-angiogenic and anti-proliferative properties, which can be further studied for the isolation of active phytochemicals for use as therapeutic agents.

## 1. Introduction

For centuries, India has been a hub for herbal medicines and a major driving force for the success of eastern medicine. Due to the high ethnic diversity, the pool of herbal medicines is enormous in India. *Parkia javanica (PJ)*, commonly called *yongchak* in Manipuri, is one of the

**Funding:** The author(s) received no specific funding for this work.

**Competing interests:** The authors have declared that no competing interests exist.

widely used medicinal plants in North-East India. *Parkia* fruits are used as home remedies for dysentery and piles or as vegetables in steamed or cooked form [1, 2]. *P. javanica* extracts are known for antioxidant activity [3, 4], and anticarcinogenic properties [4, 5].

One of the important hallmarks of malignant tumors is angiogenesis, the process of formation of new blood vessels from the native vasculature. Tumor cells grow at a rapid pace and are generally hypoxic. Thus, for delivering nutrition and oxygen to the hypoxic microenvironment, it requires synthesis of new blood vessels [6]. Tumor cells secrete VEGF that binds with VEGF receptor-1 and VEGF receptor-2, which are expressed on vascular endothelial cells and increases angiogenesis. Induction of angiogenesis by tumor cells is not only for nutrition and oxygen supply but also for ensuring successful metastasis of malignant cells. The genes responsible for production of VEGF are up regulated by oncogene expression [7, 8]. Since tumor growth and movement to distant sites is dependent on angiogenesis [9], inhibition of angiogenesis can be a therapeutic approach.

Tumor cells divide rapidly and release various growth factors in the microenvironment that promote cell proliferation [10]. Therefore, targeting these characteristics of tumor cells using natural extracts can be possible way ahead towards development of new therapeutics. A vast majority of western medicines are derived from plant-based compounds, and hence there are numerous possibilities for herbal treatment in modern era. Numerous *in vitro* studies have been conducted on putative plant-based drugs, but these studies lacked the perspective of an *in vivo* setting with its multifactorial factors and systemic approach to drug testing. Plant-based products are believed to have lesser side effects compared to modern medicines and the Indian traditional system documents the use of many such compounds that are used for treating various conditions, including cancer. Modern anti-cancer drugs have many side effects and drug resistance is a common phenomenon. Thus, targeting cancer through alternative approaches and identification of plant-derived compounds for such approaches are highly relevant. Although cell culture provides information about potential side effects, multicellular model gives us multifactorial systemic effects. So, for the compound to be safe in market use, the toxicity profiling should be done in systems with a physiological relevance [11]. Therefore, identification of inhibitors for angiogenesis and regulators for pro and anti-apoptotic proteins from natural sources holds great promise in cancer management.

Zebrafish (*Danio rerio*), a small tropical fish, serves as an excellent model for disease modeling because of its cost-effective maintenance, rapid fertility, ease of growth and amenability to large scale phytochemical screening at reasonable cost [12]. The *in-vitro* model systems provide information about potential side effects, but multicellular animal-based study provides the physiological relevance. Therefore, it is essential to determine the toxicity of compounds at physiological level and model systems like zebrafish can serve as useful *in vivo* tools to support for the selection of the safest lead molecules in the path of drug discovery process [13]. For instance, the availability of *Kdrl*:eGFP transgenic zebrafish model system is one of the important research tool for angiogenesis study [14, 15]. In the present study, we evaluated the potential effects of anti-angiogenesis, anti-proliferative and apoptotic activity of *Parkia javanica* bark and fruit extract in zebrafish embryos.

## 2. Materials and methods

### 2.1. Sample collection and extract preparation

The fresh bark and fruit of *Parkia javanica* (Lam.) Merr. were obtained from North eastern state of Tripura, India. The identification and authentication of the plant species was done by a certified botanist. The details are given in S1 Table. Briefly, *P. javanica* bark and fruits were cleaned with distilled water, then chopped into tiny pieces, and shaded dried for 3–4 weeks.

The dried materials were then ground, and 200 gm of powdered bark and fruits was mixed with 600 mL of methanol to prepare the methanolic extract of *PJ* bark (*MEPJB*) and fruits (*MEPJF*). The solution was kept in a shaker at room temperature for 48 hours followed by filtration at least 3–4 times with Whatman filterpaper-1. Then the filtered solution was concentrated by evaporating the pure methanol solvent from the solution in a rotary evaporator at 45˚C. Finally, the concentrated solution was dried at 4˚C to eliminate the methanol from the extract. The % yield of the extract was 8.896% and was stored at -20˚C [16].

## 2.2. Zebrafish husbandry and rearing

Adult zebrafish were reared in the multilinking culture system with the ambient temperature maintained around at 28.5˚C with a fixed 14/10 h day/night cycle. The fishes were kept pairwise in the breeding chambers overnight. Embryos were collected the next morning and transferred to fresh petri dishes containing E3 media and kept in incubator at 28.5˚C temperature [17]. The embryos were segregated at 10 hours post fertilization and exposed to both the compounds.

## 2.3. Determination of sub lethal concentration (LC$_{50}$)

To determine the LC$_{50}$ of both the extracts, a semi-static methodology was followed, where media was changed twice in a day along with the compound. Probit analysis was used to find the LC$_{50}$ value of both extracts. The upper limit was the concentration where all embryos died (LC$_{100}$) and lower limit was the concentration at which all embryos were survived (LC$_0$). The embryos (n = 10) were segregated and put in a 12-well micro titer plate and exposed to both the compounds at 10 hpf. The experiment was conducted in triplicates with one unexposed control group. 24 hours post exposure, embryos were observed for survivability and were imaged using a stereomicroscope (Leica S9D, Germany).

## 2.4. Evaluation of developmental toxicity of bark and fruit extracts

Developmental toxicity of *P. javanica* bark and fruit extracts were assessed by observing the embryos for five days and noting the sub lethal and sub-sub lethal end points at each day of post fertilization. Cardiotoxicity, neurotoxicity, hepatotoxicity and hatching ability were examined at 1/10$^{th}$ and 1/100$^{th}$ of sub lethal concentration. 10 embryos were used for each concentration and were imaged using a stereomicroscope (Leica S9D, Germany) and compared with control. All the experiments were performed in triplicates.

## 2.5 Assessment of anti-angiogenesis property of *P. javanica* bark and fruit extracts

To assess the anti-angiogenic property of both the extracts, transgenic zebrafish line (Tg (*Kdrl*: GFP)) where GFP protein expresses under the control of the promoter of the *kdrl* gene was used. Tg (*Kdrl*:GFP) embryos were treated with sub lethal concentration of both bark and fruit extracts (1/10$^{th}$ and 1/100$^{th}$) at 10 hpf and compared with control. All the embryos were treated with 10 μl of phenylthiourea (0.003%) to prevent pigmentation. The treated embryos were observed for intersegmental vessels (ISV) and sub intestinal veins (SIV) at 48 and 72 hpf using fluorescent microscope (Leica 4500, Germany). The analysis of image was done using ImageJ software and fluorescence intensity was compared using CTCF value.
[CTCF = Integrated Density–(Area of selected cell X Mean fluorescence of background readings)] [18].

## 2.6. Evaluation of anti-proliferative activity of *P. javanica* bark and fruit extracts

The anti-proliferative activity of both the extracts was evaluated on zebrafish AB line embryos using pH3-histone staining [20]. Phosphorylation of serine10 residue in H3 histone tail is the characteristics feature of DNA condensing during mitosis. So, phosphorylated histone H3 antibody staining is done to detect the proliferation of cells. The healthy embryos were exposed with both extracts at 10hpf and dechorionation was performed at 24 hpf, followed by fixation of the embryos in 4% PFA (paraformaldehyde). The pH3 staining was done at 6 hours post fixation. In brief, the excess PFA and the organic waste was removed and washed with 1X PBS and acetone respectively. Permeability of cells was increased by using PBST and to avoid non-specific binding, blocking solution was used. Primary antibody was introduced and incubated overnight at 4˚C. On the second day, secondary antibody was added, which is conjugated with peroxidase enzyme. Peroxidase conjugated enzyme uses hydrogen peroxide as a substrate and oxidize the DAB solution (3,3'diaminobenzene). Imaging was done using stereomicroscope (Leica S9D, Germany).

## 2.7. Assessment of apoptotic activity of *P. javanica* bark and fruit extracts

Apoptotic activity of the extracts was evaluated using acridine orange staining in $tp53^{-/-}$ mutant embryos. Acridine orange is an organic nucleic acid-specific dye that can easily cross the cellular membrane and bind with the dsDNA. After binding with dsDNA, it emits green fluorescence. The embryos were exposed to the extracts at 10hpf along with PTU (0.003%) to prevent pigmentation. Post exposure for 24 hpf, 1.5μl of acridine orange (0.5X) was added to 3 ml of E3 media. The embryos were incubated in dark for 30 minutes and post incubation embryos were washed with E3 media and observed under fluorescent microscope (Leica 4500, Germany). The analysis of image was done using ImageJ software and fluorescence intensity was calculated using CTCF value. [CTCF = Integrated Density–(Area of selected cell X Mean fluorescence of background readings)].

## 2.8. Chemicals and reagents

Acradine Orange (Sigma-Aldrich, Germany), phenyl 2-thiourea (PTU) (Sigma-Aldrich, Germany), 2',7'-dichlorodihydrofluorescein diacetate or H2DCFD (Sigma-Aldrich, Germany) Dimethylsulfoxide (DMSO) (Himedia, India Pvt, Ltd.), KCl (Merck Life Science Germany.), CaCl2(Thermo fisher scientific, India Pvt, Ltd.), NaCl (Himedia, India Pvt, Ltd.), MgSO4 (Loba Chemie India Pvt, Ltd.).

## 2.9. Statistical analysis

The differences in values between the control and crude-extract treated embryos were checked for significance by estimating the p-value using a t-test. A p value of $<0.05$ was considered to be statistically significant.

## 3. Results

### 3.1 Effects of *P. javanica* extracts on development

The $LC_{50}$ concentration of bark extract was found to be 28.66 mg/L as compared to fruit extract that showed a much higher value of 346.6 mg/L, suggesting it to be more toxic compared to that of fruit extract. In case of *P. javanica* (PJ) bark extract, no mortality was observed at a concentration below 23 mg/L ($LC_0$) whereas for PJ fruit extract $LC_0$ was below 250 mg/L.

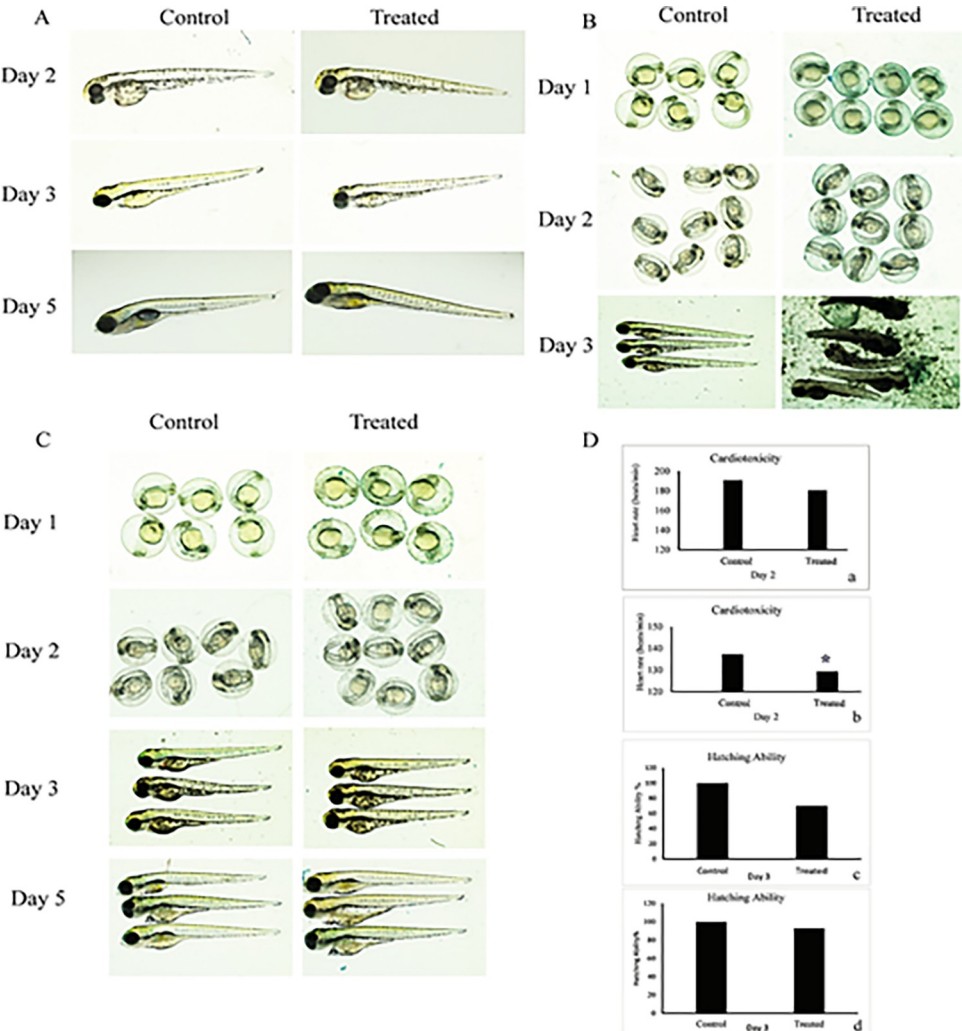

**Fig 1. Effect of *Parkia javanica* bark and fruit extracts on developmental toxicity.** (A) The *PJ* bark extracts showing morphological defects with $1/10^{th}$ concentration (2.86 ppm) of $LC_{50}$ compared to control group at different days of post fertilization. (B) & (C) represent the morphological defects of embryos treated with *PJ* fruit extract at $1/10^{th}$ and $1/100^{th}$ concentration of LC 50 *i.e.* 34.6 ppm and 3.46 ppm. (D) The cardiotoxicity of *PJ* fruit and bark in both control and treated groups was presented in (a) & (b), whereas the hatching ability was presented in (c) & (d). Asterisk indicates p-value <0.05.

The zebrafish embryos treated with $1/10^{th}$ concentration of $LC_{50}$ for PJ bark extract showed normal heartbeat, hatching ability and increased pigmentation, however subtle abnormalities were observed in pancreas at day 5 post fertilization. Intense yellow pigmentation of the intestinal bulb, along with slow development of swim bladder, was observed in the bark extract-treated embryos when compared to control (Fig 1A). However, unlike bark, *PJ* fruit extract showed developmental defects at $1/10^{th}$ (Fig 1B) and $1/100^{th}$ of sub lethal dose (Fig 1C). Exposure with $1/10^{th}$ dose of bark and fruit extracts showed an effect on the heartbeat, with a delay in the treated embryos. However, the decrease in heartbeat was statistically significant only for the fruit extract and not for the bark extract (Fig 1D). At $1/10^{th}$ of sub-lethal dose, although there was no difference in the hatching ability in both fruit and bark extract treated embryos, compared to controls (Fig 1D), the embryos showed developmental defects leading to death after 3day post fertilization (dpf). However, the $1/100^{th}$ dose of fruit extract showed no

cardiotoxicity, neurotoxicity even after hatching (3 dpf) although swim bladder abnormality was observed (Data not shown). Since the embryos treated with $1/10^{th}$ concentration of $LC_{50}$ of fruit extract could not survive beyond 3 dpf, $1/100^{th}$ concentration of the sub-lethal dose was used for all other experiments.

## 3.2 Evaluation of anti-angiogenesis property

The anti-angiogenic property of *PJ* bark and fruit extracts was evaluated in a transgenic zebra-fish model Tg(*Kdrl*:GFP) expressing GFP under the control of the promoter of kinase insert domain receptor like (*kdrl*). The antiangiogenic activity was checked at $1/10^{th}$ and $1/100^{th}$ of $LC_{50}$ concentration of *PJ* bark and fruit extract (2.86 mg/L of bark extract and 3.46 mg/L of fruit extract) as these concentrations did not show any effect on embryonic development. As shown in Fig 2A the bark extract had less effect on the formation of ISVs at 48 hpf, but at 72 hpf the treated embryos showed less branching of SIV compared to control, which is indicated by white arrow (Fig 2C). The CTCF value of treated embryos as calculated according to the signal intensity of GFP was also lesser compared to control at both 48 hpf (Fig 2B) and 72 hpf (Fig 2D). However, the difference was not statistically significant. In case of fruit extract, the treated embryos had effects on the formation of ISVs at 48 hpf (Fig 2E), the GFP intensity of the treated embryos was also less compared to the control. The CTCF value for treated embryos was found to be significantly lesser than the control embryos (Fig 2F).

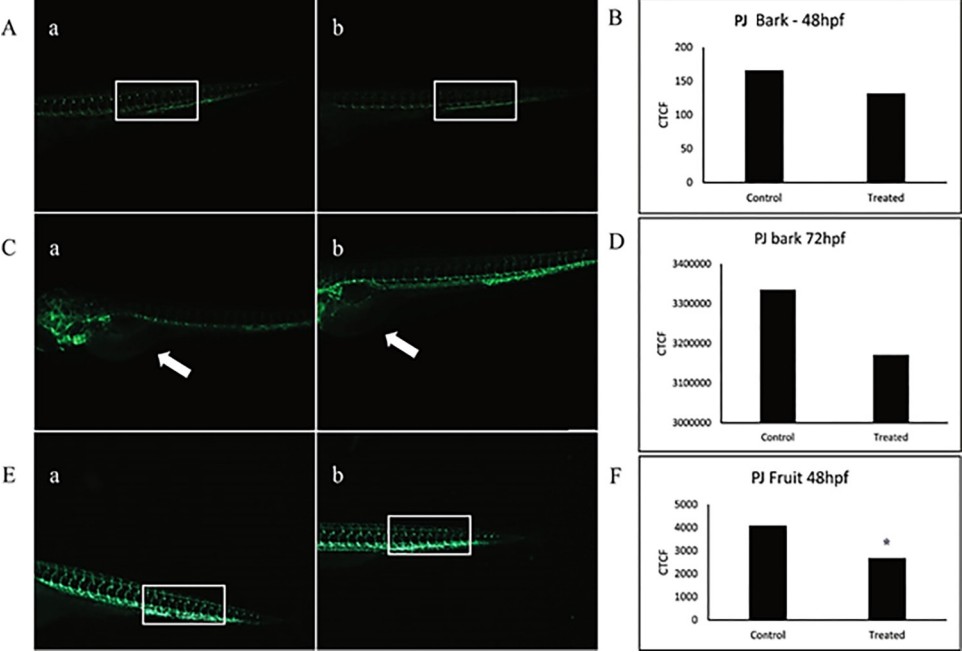

**Fig 2. Anti-angiogenic effects of *Parkia javanica* bark and fruit extract on zebrafish transgenic line Tg(*kdr*l:GFP).** (A) The anti-angiogenic activity of PJ bark extract in transgenic Tg(*kdr*l:GFP) line with 2.86 ppm at 48 hpf. (B) Graph depicting the CTCF values for PJ bark extract exposed to Tg(*kdr*l;GFP) embryos, identifying the intra segmental vessels at 48 hpf. (C) The effect of PJ bark extract (2.86 ppm) on Tg(*kdr*l:GFP) model in SIV basket at 72hpf. (D) The CTCF values for PJ bark extract exposed to Tg(*Kdr*l;GFP) embryos, identifying the intestinal basket at 72 hpf. (E) Representative images of Tg(*kdr*l:GFP) trunk ISVs (a: control; b: treated) with 3.46 ppm PJ fruit extract at 48 hpf, (F) Graph depicting the CTCF values for PJ fruit extract exposed to Tg(*Kdr*l;GFP) embryos, identifying the intra segmental vessels at 48 hpf. Asterisk indicates p-value <0.05.

### 3.3 Evaluation of anti-proliferative activity

The anti-proliferative activity of *PJ* bark and fruit extracts was assessed using wild type zebrafish embryos. The *PJ* fruit extract was found to be anti-proliferative at both 1/10th and 1/100th concentration of sub-lethal dose whereas *PJ* bark extract did not show any anti-proliferative activity (Fig 3). The cell proliferation was evaluated by counting the number of brown dots, each representing a cell positive for pH3, a proliferation marker.

### 3.4 Assessment of apoptotic activity of *Parkia javanica b*ark and fruit extracts

The apoptotic activity of *PJ* bark and fruit extracts were assessed by using tp53$^{-/-}$ mutant zebrafish embryos. Embryos treated with both the extracts showed more apoptotic cells compared to the control, indicated by an increase in the fluorescence intensity in the treated embryos (Fig 4A and 4C The corresponding CTCF values of treated *PJ* bark and fruit at 2.86 mg/L and 34.6mg/L respectively was higher compared to control, which indicates that both the extracts can induce cellular apoptosis (Fig 4B and 4D). However, statistically significant values were seen only for fruit extract.

## 4. Discussion

Zebrafish model system has proved to be a reliable *in- vivo* model system in recent progresses in drug discoveries, cancer research, and toxicity studies [19, 20]. The advantage of zebrafish over the other of animal model is due to its short lifespan, high fecundity, embryonic transparency, and low set up cost. The ability to absorb small molecules diluted in the surrounding water through their skin, gut, and gills is also another interesting feature of zebrafish [21]. The genus *Parkia* have about 34 different species and most of them are pharmacologically active [22]. The *Parkia* plants are consumable right from the inflorescence, fruits, the mature seeds, and are known to be very rich in protein and minerals. Victoria and co-researchers found that *PJ* seed extract can induce cytotoxicity in HeLa and MCF-7 cancer cell lines [23]. The aqueous and methanol extract of *PJ* fruit extract have been shown to induce cytotoxicity in

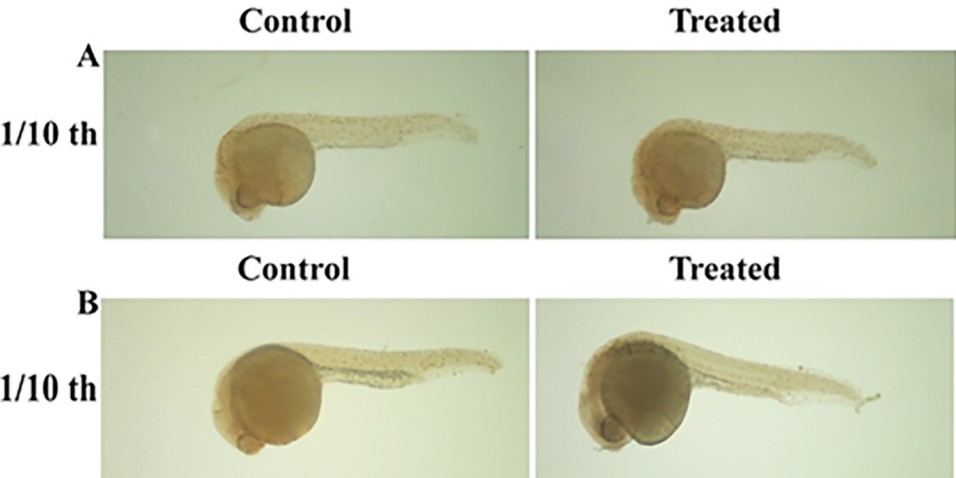

**Fig 3. Anti-proliferative effects of *Parkia javanica* bark and *f*ruit extract on zebrafish.** (A) The number of cells represented in brown dots has decreased in embryos treated with fruit extract compared to the control in 1/10th of LC$_{50}$ concentration (34.6 ppm). (B) The number of cells represented in brown dots in embryos treated with bark extract with 1/10th concentration (2.86 ppm) did not show any difference compared to control.

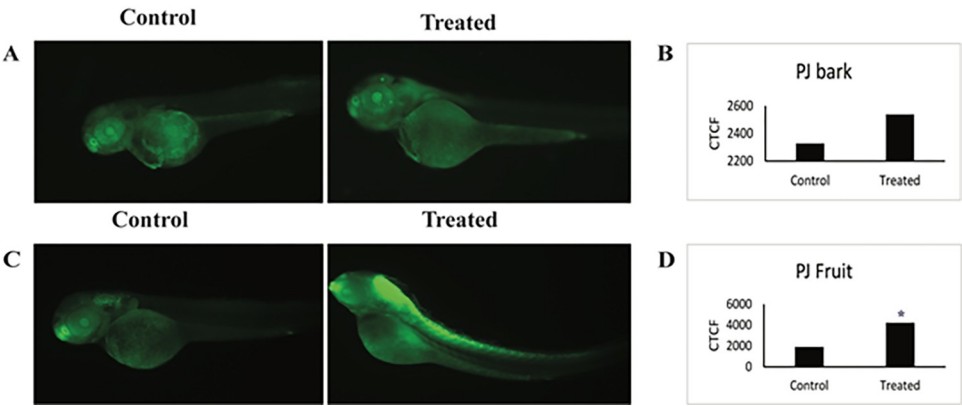

**Fig 4. Pro-apoptotic effect of *P. javanica* bark and fruit extracts.** (A) Representative images of embryos exposed to UV and treated with 2.86 ppm of bark extract showing pro-apoptosis compared to untreated embryos. (B) Graph showing the total cell fluoresce intensity exposed to 2.86 ppm PJ bark extract at 48 hpf embryos. (C) The embryos exposed to UV and treated with 3.46 ppm of fruit extract showing pro-apoptosis compared to untreated embryos. (D) Graph showing corrected total cell fluoresce of 48 hpf embryos exposed to 3.46 ppm PJ bark extract. Asterisk indicates p-value <0.05.

MDA-MB435S, sarcoma-180, A549, and AGS cell lines [24]. However, there are no toxicity reports available on *in vivo* model system. We investigated the toxicity of *PJ* extracts on zebrafish and the sub-lethal doses of bark and fruit extract were found to be 28.66mg/L and 346.7 mg/L respectively with no developmental defects except for differences in pigmentation and coloration of intestinal bulb whose etiology requires further investigation. The sub lethal doses of $LC_{50}$ of the extracts (fruit- 34.6 mg/L, 3.46 mg/L and bark- 2.86 mg/L) was used for understanding the biological activity of the compound such as developmental toxicity, and anticancer activity. The 1/100th concentration of the sub-lethal dose was found to be safer for embryos throughout the experiment.

Neovascularization or angiogenesis is the key feature in solid tumors. Therefore, characterization of anti-angiogenic property of any compounds is considered as a promising approach to cancer therapies. *Parkia speciosa* fresh pod extract (methanol) showed more than 50 percent significant inhibition of vascularization in rat aortae [22]. In our study, *P. javanica* bark extract did not show any anti-angiogenic property in zebrafish at 1/10th of sub lethal concentration. However, 1/100th sub-lethal dose of *PJ* fruit extract showed prominent anti-angiogenic activity and developmental defects such as increased pigmentation, slow development of swim bladder, cardiotoxicity, neurotoxicity.

Cancer cells have hallmark of sustained cell proliferation. *Parkia roxburghii* seed extract was shown to inhibit proliferation of B-cell hybridoma cell line [25], and HepG2 cells without influencing the normal cells [26]. Similarly, methanol extracts of *Parkia filicoidea and Parkia biglobosa* were found to exhibit anti-proliferative activities on BT-20 and T-549 (prostate cancer), SW-480 (colon cancer) and PC-3 (acute T cell leukemia Jurka) at concentration-dependent manner [27]. The phosphorylated histone H3 protein are always found in metaphasic chromosomes and by estimating the concentration phosphorylated H3 histone protein one can predict the rate of cell division in an organism [28]. In this study, immune staining was performed using anti-phosphorylated H3 histone protein antibody, which binds to the phosphorylate H3 tail during the cell division. A higher dot indicates proper chromatin condensation. In the case of *PJ* fruit extract both the sub-lethal concentration showed anti-proliferative activity indicated by the presence of lesser positive dots in treated embryos compared to control embryos.

Apoptosis is a popular target of many cancer treatment strategies. Patra and group found that the *PJ* extract significantly lowered the tumor volume and tumor weight compared to untreated group [24]. In the present study, apoptosis inducing ability of *PJ* extract was evaluated in zebrafish model and the results showed a remarkable increase in fluorescence when exposed to UV radiation suggesting that the extracts has pro-apoptotic effects. Other studies also showed pro-apoptotic activity of *PJ* seed extracts in cancer cells and bark extracts in colon cancer cell lines [26]. The findings of this study support the evidence for apoptosis inducing activity of *PJ*. In this study, crude extracts were used to evaluate the anti-proliferative and anti-angiogenic activity of *PJ*. Further investigation is required to confirm the molecular mode of action of *PJ* extracts and identification of the active ingredient. We did attempt the characterization of the PJ bark and the fruit extract through GC-MS. In case of PJ fruit, a total of 14 peaks were seen, suggesting the presence of 14 compounds and one of the them was Lupeol, a well-characterized dietary triterpene (member of the phytosterol family) with known anti-inflammatory and anti-cancer properties [29]. In case of the bark extract, a total of 15 compounds were identified, and one of the major compounds identified was D-allose, known for its anti-cancer and anti-metabolic syndrome effects [30]. The details of the GC-MS analysis are shown as S2 and S3 Tables. *Parkia javanica* fruits can be considered as important dietary components to prevent cancer development. Future studies can also be focused on isolation and characterization of phytochemicals from *PJ* fruits as potential anti-cancer compounds.

## 5. Conclusion

The study has focused on the usage of zebrafish as an effective *in-vivo* system animal model for high throughput screening of natural compounds. The study revealed that *Parkia javanica* fruit extract has anti-angiogenic, anti-proliferative activities compared to the bark extract. Future work should be focused on identifying and isolating the active phytochemicals from this crude extract and evaluating their therapeutic potential that will pave the way for the development of novel anti-cancer drugs.

## Supporting information

**S1 Table. Taxonomic identification of *Parkia javanica*.**
(DOCX)

**S2 Table. GC-MS library of *Parkia javanica* fruit extract.**
(DOCX)

**S3 Table. GC-MS library of *Parkia javanica* bark extract.**
(DOCX)

**S1 File.**
(DOCX)

## Acknowledgments

We thank NITTE (deemed to be university) for providing the research infrastructure and resources for carrying out this work.

## Author Contributions

**Conceptualization:** Anirban Chakraborty, Gunimala Chakraborty.

**Data curation:** Rasik Dhakal, Krithika Kalladka, Jeshma Ravindra.

**Formal analysis:** Krithika Kalladka, Achinta Singha, Dechamma Pandyanda Nanjappa.

**Investigation:** Rasik Dhakal, Achinta Singha, Dechamma Pandyanda Nanjappa, Rajeshwari Vittal, Gunimala Chakraborty.

**Methodology:** Samir Kumar Sil, Anirban Chakraborty, Gunimala Chakraborty.

**Supervision:** Rajeshwari Vittal, Samir Kumar Sil.

**Validation:** Achinta Singha, Jeshma Ravindra.

**Writing – original draft:** Samir Kumar Sil, Anirban Chakraborty, Gunimala Chakraborty.

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
