## [Decision Letter · Decision Letter 0]

18 Apr 2023

PONE-D-22-33455

Investigation of anti-proliferative and anti-angiogenic properties of Parkia javanica bark and fruit extracts in zebrafish

PLOS ONE

Dear author

Thank you for submitting your manuscript to PLOS ONE. After careful consideration, we feel that it has merit but does not fully meet PLOS ONE’s publication criteria as it currently stands. Therefore, we invite you to submit a revised version of the manuscript that addresses the points raised during the review process.

A rebuttal letter that responds to each point raised by the academic editor and reviewer(s). You should upload this letter as a separate file labeled 'Response to Reviewers'.A marked-up copy of your manuscript that highlights changes made to the original version. You should upload this as a separate file labeled 'Revised Manuscript with Track Changes'.An unmarked version of your revised paper without tracked changes. You should upload this as a separate file labeled 'Manuscript'

We look forward to receiving your revised manuscript.

Kind regards,

Sehrish Sadia

Academic Editor

PLOS ONE

2. Thank you for stating the following in the Acknowledgments/ Funding Section of your manuscript:

“The financial support from NITTE (Deemed to be University) through the intramural research grant.”

“The author(s) received no specific funding for this work”

Reviewers' comments:

Reviewer's Responses to Questions

**Comments to the Author**

1. Is the manuscript technically sound, and do the data support the conclusions?

Reviewer #1: Partly

Reviewer #2: Partly

2. Has the statistical analysis been performed appropriately and rigorously? 

Reviewer #1: No

Reviewer #2: No

3. Have the authors made all data underlying the findings in their manuscript fully available?

Reviewer #1: Yes

Reviewer #2: No

4. Is the manuscript presented in an intelligible fashion and written in standard English?

Reviewer #1: Yes

Reviewer #2: No

5. Review Comments to the Author

Reviewer #1: In the manuscript “Investigation of anti-proliferative and anti-angiogenic properties of Parkia javanica bark and fruit extracts in zebrafish” authors have compared the anti-proliferative and anti-angiogenic properties of bark and fruit extracts of Parkia javanica in the zebrafish model. In addition, authors have reported the efficacy of fruit extract in inhibiting the proliferation of cells compared to bark extract. Although the results are interesting, it is a preliminary study and requires extensive studies in appropriate model systems to demonstrate the anti-proliferative and anti-angiogenic properties. Furthermore, as authors have also mentioned, identification of active ingredients is lacking in this manuscript. At least authors would have determined the composition of the extract by LC-MS-MS. In light of these shortcomings, I recommend the publication of this manuscript only after including the composition analysis of the extract and addressing the following comments

Major comments

1. Did a certified botanist authenticate the plant? If so provide the authentication certificate and the photograph of the specimen used for authentication as supplemental data. Mention the full name of the plant Parkia javanica (Lam.) Merr.

2. Did authors use whole fruit for the extract preparation? Or they have separated the skin and seeds?

3. How much is the yield of methanolic extract?

4. Error bars are missing in all the bar graphs. Did authors determine the statistical significance? If so, add the information

Minor comments

Line #94. Authors have mentioned that they have prepared extracts from bark as well as fruit, but the name MEPJB, referring extract from bark, was only mentioned. Did authors have labeled the extract fruit as MEPJF? If so, include it in the methods section.

Line #129: Mention the concentration of polythiourea

Reviewer #2: [1]. The abstract does not provide much context for why the study is important or what is already known about the topic. It would be helpful to include a brief introduction that explains the prevalence and anti-proliferative and anti-angiogenic properties, and why finding new treatments is important.

[2]. The abstract does not provide information on the sample size and experimental design of the study, which is an important factor in determining the reliability of the results. Providing more information on these aspects would help readers to understand the study's rigor.

[3]. In line 52, the Manipuri name of the plant is wrong.

[4]. What about the authentication of plant material?

[5]. Voucher specimen no is missing in the MS.

[6]. A list of chemical and reagent are missing.

[7]. While using the rotary evaporator, the temperature should not be more than 45-48 oC but in the current work 60 oC was used which is not acceptable.

[8]. As mentioned in lines 97-99 “Finally, the concentrated solution was lyophilized at 4°C to eliminate the methanol..”, but lyophilization is important for removing the water traces, not the methanol.

[9]. The % yield of the extract is missing.

[10]. There is no details section of statistical analysis.

[11]. The authors did not clarify the standard drug used in the study.

[12]. It would be good if the phyto constituent of the active extract can be detected by HPLC or LC-MS or some other analytical techniques.

[13]. The presentation of the results and discussion is not well organized.

[14]. The conclusion is not effective.

[15]. There are many syntax errors, confusing sentences, and grammatical mistakes in the manuscript. The manuscript critically needs revision.

[16]. The English of the manuscript is not up to the standard of the journal.

6. PLOS authors have the option to publish the peer review history of their article (what does this mean?). If published, this will include your full peer review and any attached files.

Reviewer #1: **Yes: **SubbaRao V. Madhunapantula

Reviewer #2: No

---

## [Author Response · Author response to Decision Letter 0]

5 Jun 2023

Response to reviewer’s comments

Journal name: PLOS ONE

Manuscript ID: PONE-D-22-33455

Title: Investigation of anti-proliferative and anti-angiogenic properties of Parkia javanica bark and fruit extracts in zebrafish

All the modifications in the revised manuscript are highlighted in yellow colour.

Comments to the Author

1. Is the manuscript technically sound, and do the data support the conclusions?

Reviewer #1: Partly

Reviewer #2: Partly

2. Has the statistical analysis been performed appropriately and rigorously?

 Reviewer #1: No

Reviewer #2: No

3. Have the authors made all data underlying the findings in their manuscript fully available?

Reviewer #1: Yes

Reviewer #2: No

4. Is the manuscript presented in an intelligible fashion and written in standard English?

Reviewer #1: Yes

Reviewer #2: No

5. Review Comments to the Author

Reviewer #1: 

In the manuscript “Investigation of anti-proliferative and anti-angiogenic properties of Parkia javanica bark and fruit extracts in zebrafish” authors have compared the anti-proliferative and anti-angiogenic properties of bark and fruit extracts of Parkia javanica in the zebrafish model. In addition, authors have reported the efficacy of fruit extract in inhibiting the proliferation of cells compared to bark extract. Although the results are interesting, it is a preliminary study and requires extensive studies in appropriate model systems to demonstrate the anti-proliferative and anti-angiogenic properties. Furthermore, as authors have also mentioned, identification of active ingredients is lacking in this manuscript. At least authors would have determined the composition of the extract by LC-MS-MS. In light of these shortcomings, I recommend the publication of this manuscript only after including the composition analysis of the extract and addressing the following comments 

Response: Thank you for your valuable suggestion. The comments are well taken. We have carried out GC-MS analysis of the extracts to understand the composition of the crude extracts obtained from the fruit and the bark. However, in this manuscript the focus was on analysing the effects of the crude extracts. In the revised manuscript, the composition analysis of the crude extracts have been mentioned briefly in the discussion section (Lines 264-271) and the details are included as supplemental data (Table 2 and 3). 

Major comments

1. Did a certified botanist authenticate the plant? If so provide the authentication certificate and the photograph of the specimen used for authentication as supplemental data. Mention the full name of the plant Parkia javanica (Lam.) Merr.

Response: We thank the reviewer for this suggestion. Yes, the taxonomic identification of the plant was done by a certified botanist , Dr H.J. Chowdhary, the joint director at the Central National Herbarium, Botanical Survey of India, Shibpur, Howrah, West Bengal, India. As per the suggestion of the reviewer, the details of this identification are shown as supplemental table 1. 

2. Did authors use whole fruit for the extract preparation? Or they have separated the skin and seeds?

Response: We thank the reviewer for this comment. Yes, the whole fruit, without separating the skin and the seeds was used for the crude extract preparation. 

3. How much is the yield of methanolic extract?

Response: Thank you for your valuable suggestion. The percentage yield is now mentioned in the materials section in the revised manuscript. (Line 110) 

4. Error bars are missing in all the bar graphs. Did authors determine the statistical significance? If so, add the information

Response: We thank the reviewer for this valuable suggestion. As per the suggestions, the error bars are included in the bar graphs for all the figures. 

Minor comments

Line #94. Authors have mentioned that they have prepared extracts from bark as well as fruit, but the name MEPJB, referring extract from bark, was only mentioned. Did authors have labelled the extract fruit as MEPJF? If so, include it in the methods section.

Response: Thank you for your valuable suggestion. Corrections have been made in the revised manuscript according to the reviewer’s suggestion (Lines 105-106). 

Line #129: Mention the concentration of polythiourea

Response: Thank you for your valuable suggestion. Corrections have been made in the revised manuscript according to reviewer’s suggestion the concentration of PTU is indicated in the method section (Line 141).

Reviewer #2: 

[1]. The abstract does not provide much context for why the study is important or what is already known about the topic. It would be helpful to include a brief introduction that explains the prevalence and anti-proliferative and anti-angiogenic properties, and why finding new treatments is important.

Response: We thank the reviewer for this suggestion. We have now added a few sentences highlighting the relevance of the study (Lines 23-26). However, since the abstract has certain word limits, we couldn’t add to many information there. However, explanations on why new treatments are important are mentioned in the introduction part (Lines 72-82). 

[2]. The abstract does not provide information on the sample size and experimental design of the study, which is an important factor in determining the reliability of the results. Providing more information on these aspects would help readers to understand the study's rigor.

Response: We thank the reviewer for this suggestion. For a study of this nature, sample size is not very informative as the focus was on determining the anti-cancer properties of crude extracts of fruit and bark of a plant, Parkia javanica. However, as per the suggestion, we have now included a few sentences about the experimental design in the abstract of the revised manuscript (Lines 31-33). 

[3]. In line 52, the Manipuri name of the plant is wrong.

Response: Thank you for your valuable suggestion. Corrections in the local name of the plant have been made in the revised manuscript according to reviewer’s suggestion (Line 56)

[4]. What about the authentication of plant material?

Response: Thank you for your valuable suggestion. The details of authentication of the plant material have been added in the revised manuscript according to reviewer’s suggestion as a supplemental data and the same has been mentioned in the methods section. (Lines 101-102). 

[5]. Voucher specimen no is missing in the MS.

Response: Thank you for your valuable suggestion. The details of voucher specimen no has been added as a supplemental data in the revised manuscript.

[6]. A list of chemical and reagent are missing.

Response: Thank you for your valuable suggestion. The list of chemical and reagents used are incorporated in the materials and methods section in the revised manuscript (Lines 175-179). 

[7]. While using the rotary evaporator, the temperature should not be more than 45-48 oC but in the current work 60 oC was used which is not acceptable.

Response: Thank you for your valuable correction. The temperature for the rotary evaporator was fixed at 45oC. It was a typographical error, it has been modified in the revised manuscript. 

[8]. As mentioned in lines 97-99 “Finally, the concentrated solution was lyophilized at 4°C to eliminate the methanol..”, but lyophilization is important for removing the water traces, not the methanol.

Response: Thank you for your valuable correction. We have not lyophilized the samples. It was simply dried at 4C for completely removal of methanol. The word lyophilized have been replaced with dried in the revised manuscript (line 109).

[9]. The % yield of the extract is missing.

Response: Thank you for your valuable comment. The yield of the extract has been incorporated in the revised method section (Line 110).

[10]. There is no details section of statistical analysis.

The details of statistical significance is mentioned in the materials and methods section (Line 181-183). 

[11]. The authors did not clarify the standard drug used in the study.

We would like to clarify here that there was no standard drug used in this study. The observations were compared between the embryos treated with crude extracts and those untreated. 

[12]. It would be good if the phyto constituent of the active extract can be detected by HPLC or LC-MS or some other analytical techniques.

Response: Thank you for your valuable comment. The GC-MS composition of both the fruit and the bark extract are shown as supplementary data (Supplementary tables 2 and 3) in the revised manuscript. 

[13]. The presentation of the results and discussion is not well organized.

Response: Thank you for your valuable comment. In the revised manuscript, the sections pertaining to the presentation of the results and the discussion were rephrased for better clarity. 

[14]. The conclusion is not effective.

Response: Thank you for your valuable comment. The conclusion part has been rephrased. 

[15]. There are many syntax errors, confusing sentences, and grammatical mistakes in the manuscript. The manuscript critically needs revision.

Response: Thank you for your valuable comment. As per the reviewer’s suggestion, the entire manuscript has been revised extensively to remove the syntax errors and grammatical errors. 

[16]. The English of the manuscript is not up to the standard of the journal.

Response: Thank you for your valuable comment. The manuscript has been edited extensively to ensure that the language used is as scientific as possible and the usage of the vocabulary is as per the requirements of scientific journals. 

6. PLOS authors have the option to publish the peer review history of their article (what does this mean?). If published, this will include your full peer review and any attached files.

Do you want your identity to be public for this peer review? For information about this choice, including consent withdrawal, please see our Privacy Policy.

 Reviewer #1: Yes: SubbaRao V. Madhunapantula

Reviewer #2: No

---

## [Editor Report · Decision Letter 1]

12 Jul 2023

Investigation of anti-proliferative and anti-angiogenic properties of Parkia javanica bark and fruit extracts in zebrafish

PONE-D-22-33455R1

Dear Author

We’re pleased to inform you that your manuscript has been judged scientifically suitable for publication and will be formally accepted for publication once it meets all outstanding technical requirements.

Kind regards,

Sehrish Sadia

Academic Editor

PLOS ONE

Additional Editor Comments (optional):

Reviewers' comments:

<quillbot-extension-portal></quillbot-extension-portal>

---

## [Editor Report · Acceptance letter]

14 Jul 2023

PONE-D-22-33455R1 

Investigation of anti-proliferative and anti-angiogenic properties of *Parkia javanica* bark and fruit extracts in zebrafish 

Dear Dr. Chakraborty:

I'm pleased to inform you that your manuscript has been deemed suitable for publication in PLOS ONE. Congratulations! Your manuscript is now with our production department. 

Kind regards, 

on behalf of

Dr. Sehrish Sadia 

Academic Editor

PLOS ONE